# Diets including Animal Food Are Associated with Gastroesophageal Reflux Disease

Luciana Baroni [1,*], Chiara Bonetto [2], Irene Solinas [3], Pierfrancesco Visaggi [3], Alexey V. Galchenko [1], Lucia Mariani [3], Andrea Bottari [3], Mattia Orazzini [3], Giada Guidi [3], Christian Lambiase [3], Linda Ceccarelli [3], Massimo Bellini [3], Edoardo V. Savarino [4] and Nicola de Bortoli [3,5]

[1] Scientific Society for Vegetarian Nutrition, 30171 Venice, Italy; alexey.galchenko@scienzavegetariana.it
[2] Section of Psychiatry, Department of Neuroscience, Biomedicine and Movement Sciences, University of Verona, 37134 Verona, Italy; chiara.bonetto@univr.it
[3] Division of Gastroenterology, Department of Translational Research and New Technologies in Medicine and Surgery, University of Pisa, 56126 Pisa, Italy; i.solinas1@studenti.unipi.it (I.S.); pierfrancesco.visaggi@phd.unipi.it (P.V.); l.mariani10@studenti.unipi.it (L.M.); a.bottari@studenti.unipi.it (A.B.); m.orazzini4@studenti.unipi.it (M.O.); 27627122@studenti.unipi.it (G.G.); l.ceccarelli@ao-pisa.toscana.it (L.C.); massimo.bellini@unipi.it (M.B.); nicola.debortoli@unipi.it (N.d.B.)
[4] Division of Gastroenterology, Department of Surgery, Oncology and Gastroenterology, University of Padua, 35124 Padua, Italy; edoardo.savarino@unipd.it
[5] NUTRAFOOD, Interdepartmental Center for Nutraceutical Research and Nutrition for Health, University of Pisa, 56124 Pisa, Italy
* Correspondence: luciana.baroni@scienzavegetariana.it

**Abstract:** Gastroesophageal reflux disease (GERD) is a clinical condition with a prevalence of up to 25% in Western countries. Typical GERD symptoms include heartburn and retrosternal regurgitation. Lifestyle modifications, including diet, are considered a first-line therapeutic approach. To evaluate the impact of life habits on GERD in this cross-sectional study, we used data collected through an online survey from 1146 participants. GERD was defined according to the Montreal Consensus. For all participants, clinical and lifestyle characteristics were recorded. Overall, 723 participants (63.1%) consumed a diet including animal food (non-vegans), and 423 participants (36.9%) were vegans. The prevalence of GERD was 11% (CI 95%, 9–14%) in non-vegans and 6% (CI 95%, 4–8%) in vegans. In the multivariate analysis, after adjusting for confounding factors, subjects on a non-vegan diet were associated with a two-fold increase in the prevalence of GERD compared to vegans (OR = 1.96, CI 95%, 1.22–3.17, $p = 0.006$). BMI and smoking habits were also significantly associated with GERD. This study shows that an animal food-based diet (meat, fish, poultry, dairy, and eggs) is associated with an increased risk of GERD compared to a vegan diet. These findings might inform the lifestyle management of patients with GERD-related symptoms.

**Keywords:** gastroesophageal reflux disease; GERD; animal-based diet; vegan diet; heartburn; regurgitation; lifestyle habits

## 1. Introduction

Heartburn and acid regurgitation are common in the general population and can cause varying levels of discomfort depending on the frequency and intensity of symptoms. While occasional mild reflux episodes are usually harmless and don't interfere with daily activities, more frequent and severe reflux symptoms can lead to health problems, including erosive esophagitis [1]. According to the Montreal consensus [2], gastroesophageal reflux disease (GERD) is diagnosed when the reflux of stomach contents causes troublesome symptoms and/or complications. In questionnaire-based studies, GERD is defined by the presence of heartburn and regurgitation, regardless of their severity, occurring at least two days per week [3–5]. Based on this definition, GERD prevalence ranges from approximately 10% to 25% in Western countries, with increasing disease burden in recent decades [5].

The increasing incidence of GERD has been linked to the Western lifestyle. Factors such as cigarette smoking, alcohol consumption, a high body mass index (BMI), the presence of a hiatus hernia, inadequate sleep, and lack of aerobic exercise have all been associated with reflux symptoms or GERD [6–8]. Additionally, certain foods and drinks can trigger symptomatic reflux, such as citrus, carbonated beverages, chocolate, and other food items [9]; reducing their consumption might decrease the prevalence and severity of reflux symptoms [10]. Recent observations also suggest that following a predominantly Mediterranean diet, which is a plant-based diet, is associated with a lower risk of reflux disease [11,12].

There is some evidence suggesting that vegetarian diets may offer protection against reflux disease [13–15]. However, there are currently no studies specifically assessing the prevalence of gastroesophageal reflux symptoms in subjects with a prevalent intake of animal food compared to those with an exclusive intake of plant food.

In this study, we performed a questionnaire-based online survey conducted among individuals from the general population. The primary aim was to compare the prevalence of GERD-related symptoms between consumers of animal food compared to subjects with an exclusive intake of plant food.

## 2. Materials and Methods

### 2.1. Study Design

The INVITA study (INVestigation on ITAlians' habits and health) is a cross-sectional study conducted using an online survey, which began on 26 July 2022, with the aim of collecting data on lifestyle habits, health conditions, and diet of the general Italian population. Participants were recruited online, on a voluntary basis, by the advertising of the survey access link through social media and newsletters. The exclusion criteria were being younger than 18 years, pregnancy or breastfeeding, and restrictive plant-based diets (macrobiotic, fruitarian, raw food, and hygienist). The survey ensured anonymity, and informed consent was obtained from all participants. The online questionnaires were hosted by the Scientific Society for Vegetarian Nutrition (a non-profit Italian organization) as a dedicated application on the domain studioinvita.it and could be accessed using computers, tablets, or smartphones. The collected data were downloaded and managed by responsible data management personnel who had no means of identifying study participants. This study was approved by the Bioethical Committee of the University of Pisa, Italy (Prot. N. 0116339/2021, approval date 29 September 2021).

### 2.2. Study Instruments

Sociodemographic characteristics were collected using an ad-hoc form, and included gender, age, marital status, living condition, education level, and occupation. Collected lifestyle habits included self-reported height and weight, diet (consumption of meat, fish, poultry, dairy, and eggs), alcohol consumption per month (1 alcohol unit, AU = 12 g of pure alcohol, which corresponds to an average 330 cc of beer, or 125 cc of wine, or 80 cc of vermouth, or 40 cc of liquor. 'At risk' consumption was defined as >60 AUs for males and >30 AUs for females [16,17] and currently a smoker (yes/no).

The dietary pattern (non-vegan or vegan) was defined by considering 'non-vegans' as those who consumed at least one food among meat, fish, poultry, dairy and eggs, and 'vegans' as those who did not consume meat, fish, poultry, dairy, and eggs. The BMI was calculated by dividing weight in kilograms by height in squared meters.

GERD was assessed by evaluating the presence of heartburn, regurgitation, and chest pain according to the Montreal consensus, and patients were categorized as having (GERD+) or not having (GERD−) GERD [2]. Symptoms were considered to be GERD-related when they occurred at least two or more times per week in the previous 30 days. An ad-hoc question regarding medications (antiacids, histamine 2 blockers, and/or proton pump inhibitors) was also asked. Participants taking such medications were classified as GERD+, regardless of GERD-related symptoms.

*2.3. Statistical Analysis*

Categorical variables were described as absolute numbers and percentages; continuous variables were summarized as means and standard deviations (SDs). Comparisons between groups were performed with the Chi-square test in the case of categorical variables, and with the *t*-test in the case of continuous variables. Subsequently, a series of univariate logistic regression models with having GERD+ as the dependent variable and each characteristic (gender, age, marital status, living condition, education, occupation, BMI, dietary pattern, alcohol consumption, and smoking) as the independent variable were estimated (unadjusted ORs). Those characteristics that resulted in an association (at $p < 0.05$) with GERD positivity entered the multivariate logistic regression model, thus giving adjusted ORs. All tests were bilateral, with a significance level of 0.05. Analyses were performed using Stata 17 for Windows.

**3. Results**

At the time the data were extracted (16 May 2023), 4352 subjects completed sociodemographics and life habits questionnaires. Out of them, 1146 (26.3%) completed the GERD survey, and they constituted the study sample. Participants in the GERD survey giving information about medications were 962 (184 missing). In this sub-sample, people taking antiacids, histamine 2 blockers, and/or proton pump inhibitors (PPI) were 19. Out of them, 6 were GERD+ and 13, who were GERD−, were classified as GERD+. Among the 19 subjects, 15 declared to take PPI. Out of them, 5 were GERD+ and 10, who were GERD−, were classified as GERD+.

By considering sociodemographic characteristics and life habits, completers, with respect to non-completers, were slightly older (37.1, SD 12.0 vs. 35.2 SD 11.8; *t* test, $p < 0.001$), less often living with friends/other relatives/others (6.7% vs. 9.8%; Chi-square test, $p = 0.008$), more often vegans (36.9% vs. 31.7%; Chi-square test, $p = 0.001$) and, finally, less often alcohol consumers belonging to the category 'at risk' (5.0% vs. 8.2%; Chi-square test, $p < 0.001$).

The description of the study sample is given in Table 1 (a and b). More than 90% were females; the mean age was 37 years (SD 12). The majority of the subjects had a degree or a post-degree (66.5%) and were employed (70.9%). More than 60% were married or co-habitants. By considering life habits, the mean BMI was 22.2 (SD 3.8), 37% were vegans, 5% belonged to the 'at risk' alcohol consumption per month, and 91% did not smoke.

The prevalence of gastroesophageal reflux and dyspepsia (GERD+) in the study sample was 9% (95%CI 8–11%), with a significant difference with dietary patterns [non-vegans 11% (95%CI 9–14%) vs. vegans 6% (95%CI 4–8%)]. By comparing all socio-demographic characteristics and life habits, only the BMI (GERD+ 24.1 SD 5.4 vs. GERD− 22.0 SD 3.5; $p < 0.001$ *t* test), the vegan dietary pattern (GERD+ 23.6% vs. GERD− 38.3%; $p = 0.003$ Chi-square test), and the smoking habit (GERD+ 16.2% vs. GERD− 8.5%; $p = 0.009$ Chi-square test) significantly differed between the two groups.

Table 2 shows the unadjusted OR for each sociodemographic characteristic and life habit for GERD+ participants.

BMI, dietary pattern, and smoking ($p < 0.05$) entered the multivariate logistic regression model by giving adjusted ORs (Table 3). A non-vegan dietary pattern (adj-OR = 1.96, $p = 0.006$), a higher BMI (adj-OR = 1.11, $p < 0.001$), and smoking (adj-OR = 2.09, $p < 0.001$) were significantly associated with having reflux and dyspepsia.

**Table 1.** Sociodemographic characteristics (a) and life habits (b) of the overall sample and of GERD+ and GERD− participants (n = 1146).

| (a) Sociodemographic Characteristics | Overall Sample n = 1146 | GERD− n = 1040 (90.8%) | GERD+ n = 106 (9.2%) | *p*-Value |
|---|---|---|---|---|
| Gender, n (%) | | | | 0.509 |
| Male | 83 (7.2%) | 77 (7.4%) | 6 (6.7%) | Chi-square |
| Female | 1063 (92.8%) | 963 (92.6%) | 100 (94.3%) | |
| Age, mean (SD) | 37.1 (12.0) | 37.0 (11.9) | 37.7 (12.9) | 0.583 *t* test |
| BMI, mean (SD) | 22.2 (3.8) | 22.0 (3.5) | 24.1 (5.4) | <0.001 *t* test |
| Marital status, n (%) | | | | 0.888 |
| Married/Cohabitant | 702 (61.3%) | 635 (61.1%) | 67 (63.2%) | Chi-square |
| Separated/Divorced/Widowed | 41 (3.6%) | 37 (3.6%) | 4 (3.8%) | |
| Single | 403 (35.2%) | 368 (35.4%) | 35 (33.0%) | |
| Living condition, n (%) | | | | 0.315 |
| Family of origin | 208 (18.2%) | 184 (17.7%) | 24 (22.6%) | Chi-square |
| Partner and/or children | 707 (61.7%) | 642 (61.7%) | 65 (61.3%) | |
| Alone | 154 (13.4%) | 145 (13.9%) | 9 (8.5%) | |
| Friends/Other relatives/Others | 77 (6.7%) | 69 (6.6%) | 8 (7.5%) | |
| Education, n (%) | | | | 0.151 |
| Professional qualification | 45 (3.9%) | 40 (3.8%) | 5 (4.7%) | Chi-square |
| Diploma | 339 (29.6%) | 298 (28.7%) | 41 (38.7%) | |
| Degree | 599 (52.3%) | 553 (53.2%) | 46 (43.4%) | |
| Post-degree | 163 (14.2%) | 149 (14.3%) | 14 (13.2%) | |
| Occupation, n (%) | | | | 0.239 |
| Employed | 812 (70.9%) | 744 (71.5%) | 68 (64.1%) | Chi-square |
| Housewife/Student/Retired | 271 (23.6%) | 239 (23.0%) | 32 (30.2%) | |
| Unemployed | 63 (5.5%) | 57 (5.5%) | 6 (5.7%) | |
| (b) Life Habits | Overall Sample n = 1146 | GERD− n = 1040 (90.8%) | GERD+ n = 106 (9.2%) | *p*-Value |
| Dietary pattern, n (%) | | | | 0.003 |
| Non-vegan | 723 (63.1%) | 642 (61.7%) | 81 (76.4%) | Chi-square |
| Vegan | 423 (36.9%) | 398 (38.3%) | 25 (23.6%) | |
| Monthly alcohol consumption, n (%) | 33 missing | 31 missing | 2 missing | 0.870 |
| No consumption | 243 (21.8%) | 222 (22.0%) | 21 (20.2%) | Chi-square |
| Low/Moderate [1] | 814 (73.1%) | 737 (73.0%) | 77 (74.0%) | |
| At risk [2] | 56 (5.0%) | 50 (5.0%) | 6 (5.8%) | |
| Currently smoking, n (%) | 5 missing | 4 missing | 1 missing | 0.009 |
| No | 1036 (90.8%) | 948 (91.5%) | 88 (83.8%) | Chi-square |
| Yes | 105 (9.2%) | 88 (8.5%) | 17 (16.2%) | |

[1] ≤60 alcohol units for males; ≤30 alcohol units for females [17]; [2] >60 alcohol units for males; >30 alcohol units for females [17].

**Table 2.** Univariate logistic models for GERD+ participants: unadjusted ORs (n = 1146).

| Independent Variable | OR (Unadjusted) | 95% CI | *p*-Value |
|---|---|---|---|
| Gender | | | |
| Male | Ref. | - | - |
| Female | 1.33 | 0.57–3.14 | 0.511 |
| Age | 1.00 | 0.99–1.02 | 0.582 |
| BMI | 1.12 | 1.07–1.17 | <0.001 |
| Marital status | | | |
| Married/Cohabitant | Ref. | - | - |
| Separated/Divorced/Widowed | 1.02 | 0.35–2.96 | 0.964 |
| Single | 0.90 | 0.59–1.38 | 0.635 |
| Living condition | | | |
| Family of origin | Ref. | - | - |
| Partner and/or children | 0.78 | 0.47–1.27 | 0.317 |
| Alone | 0.48 | 0.21–1.05 | 0.068 |
| Friends/Other relatives/Others | 0.89 | 0.38–2.07 | 0.785 |
| Education | | | |
| Professional qualification | Ref. | - | - |
| Diploma | 1.10 | 0.41–2.95 | 0.849 |
| Degree | 0.66 | 0.25–1.77 | 0.414 |
| Post-degree | 0.75 | 0.25–2.21 | 0.604 |
| Occupation | | | |
| Employed | Ref. | - | - |
| Housewife/Student/Retired | 1.46 | 0.94–2.28 | 0.092 |
| Unemployed | 1.15 | 0.48–2.77 | 0.752 |
| Marital status | | | |
| Married/Cohabitant | Ref. | - | - |
| Separated/Divorced/Widowed | 1.02 | 0.35–2.96 | 0.964 |
| Single | 0.90 | 0.59–1.38 | 0.635 |
| Dietary pattern | | | |
| Vegan | Ref. | - | - |
| Non-vegan | 2.01 | 1.26–3.20 | 0.003 |
| Monthly alcohol consumption | | | |
| No consumption | Ref. | - | - |
| Low/Moderate [1] | 1.10 | 0.67–1.83 | 0.700 |
| At risk [2] | 1.27 | 0.49–3.31 | 0.626 |
| Currently smoking | | | |
| No | Ref. | - | - |
| Yes | 2.08 | 1.18–3.65 | 0.011 |

[1] ≤60 alcohol units for males; ≤30 alcohol units for females [17]. [2] >60 alcohol units for males; >30 alcohol units for females [17].

The association between each item for detecting GERD and the dietary pattern is shown in Table 4. A significantly higher percentage of non-vegans experienced, in the previous month, a burning sensation in the center of the chest going up from the stomach to the neck at least two times per week (17.3% vs. 11.8%, *p* = 0.013), the sensation of liquid rising up in the throat or leaning forward at least two times per week (17.6% vs. 10.9%, *p* = 0.002), and a feeling of slow or difficult digestion with a sense of bloating after a meal more than two times per week (45.5% vs. 35.5%, *p* < 0.001).

**Table 3.** Multivariate logistic model for GERD+ participants: adjusted ORs (only independent variables significantly associated at *p* < 0.05 in univariate logistic regression models entered the multivariate logistic regression model).

| Independent Variables | OR (Adjusted) | 95% CI | *p*-Value |
|---|---|---|---|
| BMI | 1.11 | 1.07–1.17 | <0.001 |
| Dietary pattern | | | |
| Vegan | Ref. | - | - |
| Non-vegan | 1.96 | 1.22–3.17 | 0.006 |
| Currently smoking | | | |
| No | Ref. | - | - |
| Yes | 2.09 | 1.18–3.71 | <0.001 |
| Number of observations | 1141 | | |
| LR test, *p*-value | Chi2(3) = 37.74, *p* < 0.001 | | |
| Hosmer–Lemeshow goodness-of-fit (10 groups) Chi2(df), *p*-value | Chi2(8) = 14.97, *p* = 0.060 | | |
| Pearson goodness-of-fit Number of covariate patterns Chi2(df), *p*-value | 1141 Chi2(804) = 875.91, *p* = 0.039 | | |
| Area under ROC curve | 0.652 | | |

**Table 4.** Association between dietary pattern and GERD status (total score and items) (n = 1146).

| | Dietary Pattern | | *p*-Value Chi-Square |
|---|---|---|---|
| GERD Status | Non-Vegan n (%) | Vegan n (%) | |
| GERD− | 642 (88.8%) | 398 (94.1%) | 0.003 |
| GERD+ | 81 (11.2%) | 25 (5.9%) | |
| Q1 In the past 30 days, have you had a burning sensation in the center of your chest going up from your stomach to your neck at least 2 times a week or more? | | | |
| No | 598 (82.7%) | 373 (88.2%) | 0.013 |
| Yes | 125 (17.3%) | 50 (11.8%) | |
| Q2 In the past 30 days, have you had the sensation of liquid rising up in your throat or leaning forward at least 2 times a week or more? | | | |
| No | 596 (82.4%) | 377 (89.1%) | 0.002 |
| Yes | 127 (17.6%) | 46 (10.9%) | |
| Q3 In the past 30 days, have you felt the sensation of heaviness or pain in the center of your chest at least 2 times a week or more? | | | |
| No | 629 (87.0%) | 377 (89.1%) | 0.289 |
| Yes | 94 (13.0%) | 46 (10.9%) | |
| Q4 In the past 30 days, have you felt a feeling of slow or difficult digestion with a sense of bloating after a meal more than 2 times a week? | | | |
| No | 394 (54.5%) | 273 (64.5%) | <0.001 |
| Yes | 329 (45.5%) | 150 (35.5%) | |
| Q5 In the past 30 days, did you feel pain in the 'pit of your stomach' (centrally just below the ribs) more than 2 times a week? | | | |
| No | 594 (82.2%) | 350 (82.7%) | 0.802 |
| Yes | 129 (17.8%) | 73 (17.3%) | |

## 4. Discussion

In this study, we used anonymized data collected using an online survey to compare the prevalence of typical GERD-related symptoms between consumers of animal food and subjects with an exclusive intake of plant food. Animal food-based diets were associated with a two-fold increase in the prevalence of typical GERD-related symptoms, compared to vegan diets.

GERD has been identified as a major health concern, particularly in Western societies [18]. Patients with GERD report symptoms that have a significant impact on their quality of life, causing increased levels of anxiety, stress and visceral hypersensitivity [19]. Moreover, it has been established that symptomatic gastroesophageal reflux is the leading risk factor for esophageal adenocarcinoma, a cancer with a rapidly increasing incidence and a high mortality rate [20].

GERD therapy is based on acid suppression and mucosal protectant medical devices [21]. In the United States alone, GERD-related direct and indirect costs account for approximately \$15–20 billion [22], and 80% of this amount is due to drug treatments [23].

In line with the trend of proton pump inhibitor deprescription in GERD management [24], it is important to identify and reduce any modifiable risk factor of the disease. Several lifestyle factors have been associated with GERD [10]; however, beneficial effects have only been documented for weight loss and tobacco smoking cessation in obese patients and smokers, respectively, and for avoiding late evening meals and elevating the head of the bed for nocturnal GERD [8,21].

Previous data suggested that dietary changes can potentially reduce the risk of GERD. Despite conflicting evidence [25], it is widely accepted that certain foods may exacerbate typical GERD symptoms. In addition to components in food that are known to trigger GERD, certain dietary patterns and eating habits have also been linked to GERD [26,27]. For instance, a Western diet has been positively correlated with the risk of GERD, regardless of other factors [11]. In the last decades, population-based studies have shown that BMI and smoking may be considered important risk factors for GERD symptoms [19,28,29], as well as for the development of Barrett esophagus and esophageal adenocarcinoma [30]. It has also been suggested that certain foods (chocolate, peppermint, tomato, and tomato sauce) may be related to GERD symptoms, but evidence to support this information is still lacking [21]. Few studies have evaluated the effect of a diet including animal food in individuals with typical GERD-related symptoms.

Previous research regarding the impact of meat consumption on gastroesophageal reflux and its clinical implications have yielded inconsistent results. Using a food frequency questionnaire, O'Doherty et al. found that meat intake had no correlation with reflux symptoms, reflux esophagitis, or esophageal adenocarcinoma [31]. Furthermore, a large monozygotic co-twin study indicated no association between meat intake and the emergence of gastroesophageal reflux [32].

In contrast, an examination of Uygur and Han Chinese revealed that a diet high in meat increased the risk of GERD [33]. Additionally, a study performed on hospital employees showed that the number of meat servings consumed per day was associated with reflux esophagitis, although not always with reflux symptoms [34].

Our team in Pisa conducted a pathophysiologic study that evaluated the first postprandial hours after three meals. Impedance and pH analysis revealed that symptoms and acid reflux events were more prevalent following a meal with animal food compared to a meal with plant food [35]. A further possible explanation of our findings might be the increase of the proximal gastric acid secretion in the postprandial period due to the phenomenon of acid pockets present in all subjects, not only in those suffering from GERD [36]. Nevertheless, the increased amount of acid in the pocket and the increased acidity of the reflux events might be related to a greater amount of saturated fats in animal food compared to plant ones [37].

Accordingly, Van Boxel et al. demonstrated that GERD patients experienced a notable increase in heartburn and nausea during a lipid infusion, which can be attributed to

an increase in chylomicron production and secretion, which may trigger the release of cholecystokinin, a stimulant of vagal afferences [38]. Moreover, a study by Fox et al. showed that a high-fat diet, compared to a low-fat diet, had effects on esophageal acid exposure and, above all, was associated with the presence of typical reflux-related symptoms [39].

From a pathophysiological point of view, it appears clear that animal foods can represent a source of different compounds capable of increasing the reflux burden or even the perception of typical reflux-related symptoms.

On the other hand, there is strong evidence supporting a protective effect of an exclusive plant-based diet in reducing GERD symptoms. Bhatia et al., who conducted a survey of volunteers and patients in urban areas and slums of India, described a positive relationship between the consumption of non-vegetarian diets and reflux symptoms [13]. Similarly, Jung et al. reported a decreased risk of reflux esophagitis in vegetarian Buddhist priests, even when their BMI was elevated [14].

In line with this, vegetarian diets are known to be high in dietary fiber, which have been linked to a reduction in reflux symptoms [40–43]. Although the precise mechanism for this is still unclear, it has been suggested that dietary fiber may scavenge nitrites found in the stomach, thus decreasing the availability of nitric oxide synthesis [42]. This could potentially reduce nitric oxide concentration in the gastroesophageal junction and inhibit reflux [42].

The aim of our study was to compare the prevalence of typical GERD-related symptoms between consumers of animal food and subjects with an exclusive intake of plant food. Important risk factors such animal food-based diets, tobacco smoking, and higher BMI were associated with a higher prevalence of GERD-related symptoms (GERD+).

Our study showed that any typical GERD-related symptoms perceived more than two days per week occurred more frequently in non-vegan participants than in vegans. The protective effect of an exclusive plant-based diet on symptomatic reflux, as shown by multivariable analysis, was independent from confounding factors, such as BMI and smoking.

This study has several strengths. It involved a substantial number of participants (more than 1000). The use of an online survey facilitated data collection, making it convenient for participants to respond and reducing the chances of missing data. This study defined the prevalence of GERD-related symptoms according to the Montreal consensus [1] and when they occurred more than two times per week. Moreover, in our study, the number of vegans, although overrepresented with respect to their proportion in the Italian population, allowed us to have a dimension suitable for serving as a comparison sample (723 non-vegan vs. 423 vegans, 63.1% vs. 36.9%).

This study has also some limitations. The first limitation concerns the possibility of a selection bias. Despite the large sample size of participants to the INVITA study (n = 4352), the percentage of those who completed the GERD survey was low (26.3%). The comparison between those who completed GERD survey (n = 1146) and those who did not complete it (n = 3206) showed that completers were slightly older, less often living with friends/other relatives/others, more often vegans, and, finally, less often alcohol consumers belonging to the category 'at risk'. Moreover, the comparison between the whole INVITA sample (n = 4352) and the general Italian population (≥18 years) showed that there are differences in some characteristics like gender and overweight prevalence, age, education level, smoking habits, and 'at risk' alcohol consumption prevalence. The vegan dietary pattern, as mentioned above, is overrepresented [44,45]. The second limitation concerns the cross-sectional design, which does not permit the identification of causal relationships. The third limitation refers to data collection, which relies on self-reported data, which is conducive to a possible recall bias and a biased interpretation of the questions. In this study, GERD was defined based on the presence of typical symptoms. In this regard, both versions of the Lyon consensus [46,47] suggest that typical symptoms are associated with a high likelihood of having objective GERD, corroborating the use of a short treatment with PPIs in primary care. Finally, the study was conducted in Italy, which may limit the generalizability of the findings to other countries.

## 5. Conclusions

In summary, our results show that a diet including animal food (meat, fish, poultry, dairy, and eggs) is associated with troublesome reflux-related symptoms. The results of the multivariate analysis indicated a positive association between regular consumption of animal food and typical GERD-related symptoms. Our findings support that diets including animal food are associated, also after controlling for other risk factors, with a significant increase in the prevalence of gastroesophageal reflux (GERD).

These data, obtained through an observational study design, are further corroborated by the fact that the survey population featured a higher proportion of vegan subjects than found in the general population. Our findings indicated that it is quite important for general practitioners to advise patients with GERD-related symptoms to modify their dietary practices, beginning by decreasing the proportion of animal foods in their diet, in order to reduce the need for medication and improve symptom perception.

In fact, there may be various mechanisms influencing the potential triggering effects of a diet including animal food versus a plant-only diet, and this field of research requires further investigation.

**Author Contributions:** Conceptualization, L.B. and N.d.B.; methodology, C.B.; software, C.B.; validation, I.S., P.V. and A.V.G.; formal analysis, M.O., A.B. and C.B.; investigation, L.B. and A.V.G.; data curation, M.O., L.C. and M.B.; writing—original draft preparation, L.B., C.B. and N.d.B.; writing—review and editing, M.O., A.B. and L.C.; visualization, P.V., E.V.S., L.M., C.L. and G.G.; supervision, L.B. and G.G. All authors have read and agreed to the published version of the manuscript.

**Funding:** This research received no external funding.

**Institutional Review Board Statement:** This study was conducted in accordance with the Declaration of Helsinki and approved by the Bioethical Committee of the University of Pisa, (Prot. N. 0116339/2021, approval date 29 September 2021).

**Informed Consent Statement:** Informed consent was obtained from all subjects involved in the study.

**Data Availability Statement:** The data presented in this study are available on request from the corresponding author. The data are not publicly available due to privacy.

**Acknowledgments:** The authors wish to thank all the participants to the INVITA study and the Scientific Society for Vegetarian Nutrition of Italy for providing technical support to the survey.

**Conflicts of Interest:** The authors declare no conflict of interest.

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
