# Peer review of "Diets including Animal Food Are Associated with Gastroesophageal Reflux Disease"

_ejihpe, doi:10.3390/ejihpe13120189_

Round 1

Reviewer 1 Report

Comments and Suggestions for Authors

I thank the authors for the possibility of reviewing this very interesting study.

The study shows how nutrition affects in important way on different aspects of our lives and in particular the origin of some diseases. Specifically, the study is well conducted, the methodology is correct, and the study population is ample and sufficient for obtaining statistically significant results. The only note I do concerns GERD's diagnosis. In the study, the GERD was actually evaluated only indirectly through a questionnaire. In reality, the GERD is a very complex entity sometimes difficult to diagnose correctly. In fact, sometimes patients positive for questionnaires have a non-pathological Demester score to 24 hours phmetry and impedancemetry. This is because other anomalies of esophageal motility, for example, can mimic this symptom. In this sense, I think it would be advisable to discuss this aspect more importantly in the discussion section. For the rest, I think this study can truly bring something new into the current scientific panorama.

Author Response

I thank the authors for the possibility of reviewing this very interesting study.

The study shows how nutrition affects in many ways different aspects of our lives and in particular the origin of some diseases. Specifically, the study is well conducted, the methodology is correct, and the study population is ample and sufficient for obtaining statistically significant results. The only note I do concerns GERD's diagnosis. In the study, the GERD was actually evaluated only indirectly through a questionnaire. In reality, the GERD is a very complex entity sometimes difficult to diagnose correctly. In fact, sometimes patients positive for questionnaires have a non-pathological Demester score to 24 hours phmetry and impedancemetry. This is because other anomalies of esophageal motility, for example, can mimic this symptom. In this sense, I think it would be advisable to discuss this aspect more importantly in the discussion section. For the rest, I think this study can truly bring something new into the current scientific panorama.

Answer: We thank the Reviewer very much for this comment. We totally agree regarding the GERD diagnosis. We tried to emphasize in the paper that the evaluation of GERD related symptoms is the most suitable method to estimate the prevalence of GERD in large population-based study. We included in the GERD diagnosis only patients who described heartburn and regurgitation that are the typical GERD-related symptom. Both versions of Lyon Consensus (version 1.0 published in 2018 and version 2.0 on-line from a couple of weeks) suggest that heartburn and regurgitation are associated with high likelihood of having GERD and the typical clinical presentation that can be managed with a short trial of PPIs in primary care. We commented this on the limitation section.

Reviewer 2 Report

Comments and Suggestions for Authors

Good work by the authors. Minor comments:

1. Line 72: were the participants of the survey verified by any method to see if they fulfilled the inclusion and exclusion criteria?

2. Line 116: Why did only 26% of them fill the GERD survey?

3. Table 1: Dietary pattern analysis: Was there a particular reason the fisher test was used instead of the chi-squared test? Could you also mention the same results using the chi-squared test?

4. Overall, great work.

Author Response

Good work by the authors.

We thank the Reviewer for this comment.

Minor comments:

  1. Line 72: were the participants of the survey verified by any method to see if they fulfilled the inclusion and exclusion criteria?

Answer: The INVITA study was conducted using an online survey, so no direct method could be used to verify the truthfulness of data given by participants. All information were collected by self-assessments. Some questionnaires were mandatory, other ones optional. By considering the exclusion criteria, age of birth, state of pregnancy and breastfeeding, and being macrobiotic, fruitarian, raw food dieter or hygienist were all questions included in one of the mandatory questionnaires, whose compilation gave access to the other optional ones.

  1. Line 116: Why did only 26% of them fill the GERD survey?

Answer: The INVITA study was conducted to collect a great mass of data on life habits (including diet) and physical and mental health of the Italian general population. Fundamental aspects such as socio-demographic characteristics, physical activity, suffering from the most common chronic diseases (cardiovascular events, hypertension, cancer, etc.), diet, alcohol consumption, smoking habits and female sexual health (menopause, pelvic pain, abortions, etc.) were set as mandatory assessments for participants. Many other aspects such as depression, anxiety, stress, family history for chronic diseases, quality of life, sleep problems, intestinal health, gastroesophageal reflux and so on were set as non-mandatory assessments (based on a voluntary choice to give information) in order to conduct in-depth analysis regarding specific physical and mental aspects. In the Results, comparisons between participants and non-participants in GERD survey on socio-demographic characteristics, diet, alcohol consumption and smoking habits (that is all the variables involved in the analyses performed in the paper) have been detailed.

  1. Table 1: Dietary pattern analysis: Was there a particular reason the fisher test was used instead of the chi-squared test? Could you also mention the same results using the chi-squared test?

Answer: Fisher’s exact test is commonly applied to 2x2 matrices. However, following the Reviewer’s suggestion, we applied the Chi-square test to all the associations in the paper (both tables and text). No result changed. In detail, the Chi-square test was introduced in the Statistical analysis paragraph, the Results section, Table 1 and Table 4.

  1. Overall, great work.

We really appreciate this comment.

Reviewer 3 Report

Comments and Suggestions for Authors

Well written manuscript. Clear presentation of the results. However, I would like to see data of patients with PPI in table 2 or new table only for this cohort ofpatients. 

Author Response

Well written manuscript. Clear presentation of the results.

We thank the Reviewer for these comments.

However, I would like to see data of patients with PPI in table 2 or new table only for this cohort of patients. 

Answer: Participants in the GERD survey giving information about medications were 962 (184 missing). In this sub-sample, subjects who declared to take antiacids, histamine 2 blockers and/or proton pump inhibitors were 19, of whom 15 took PPI. Due to the very small sample size, the application of statistical test to compare this sub-group with other groups and the estimation of logistic regression models could be not appropriate. We are aware that this sub-group could be of interest, but its sample size is negligible. In the Results we give a brief description as follows:

‘Participants in the GERD survey giving information about medications were 962 (184 missing). In this sub-sample, people taking antiacids, histamine 2 blockers and/or proton pump inhibitors (PPI) were 19. Out of them, 6 were GERD+ and 13, who were GERD-, were classified as GERD+. Among the 19 subjects, 15 declared to take PPI. Out of them, 5 were GERD+ and 10, who were GERD-, were classified as GERD+.’